# The Diabetes Team Dynamics Unraveled: A Qualitative Study

**Eefje Van Nuland** [1,†], **Irina Dumitrescu** [2,†], **Kristien Scheepmans** [2], **Louis Paquay** [2], **Ellen De Wandeler** [1] **and Kristel De Vliegher** [2,*]

1. Working Group Diabetes of the NVKVV, 1000 Brussels, Belgium; eefje.vannuland@gmail.com (E.V.N.); e.dewandeler@nvkvv.be (E.D.W.)
2. Nursing Department, White & Yellow Cross of Flanders, 1000 Brussels, Belgium; irina.dumitrescu@vlaanderen.wgk.be (I.D.); kristien.scheepmans@vlaanderen.wgk.be (K.S.); louis.paquay@vlaanderen.wgk.be (L.P.)
* Correspondence: kristel.de.vliegher@vlaanderen.wgk.be
† These authors contributed equally to this work.

**Abstract:** Background: Diabetes is a complex disease requiring a multidisciplinary approach. However, the dynamics of this collaboration and the involvement of healthcare providers remain unclear. **Aim(s):** To explore the composition, the division of roles/tasks, and the collaboration in a diabetes team. **Methods:** A qualitative, explorative study with six focus groups was conducted, of which four focus groups were with healthcare providers ($n = 34$) and two with informal caregivers and persons with diabetes ($n = 13$). In addition, two in-depth interviews with doctors were performed. An iterative process of data analysis took place, guided by the Qualitative Analysis Guide of Leuven (QUAGOL). **Results:** All participants confirm the importance of patient empowerment and the fact that the person with diabetes should have a central role within the team. However, this has not been achieved yet. This research gives a clear insight into the dynamics of a diabetes team. Roles and tasks are allocated according to the specific expertise and knowledge of the different healthcare providers. Interprofessional collaboration is the ultimate goal. However, the diabetes team is often formed ad hoc depending on the needs of the person with diabetes and the preferences for collaboration of the healthcare providers. Furthermore, this study revealed some important bottlenecks with regard to the knowledge of healthcare providers, persons with diabetes and their informal caregivers, the regulation and reimbursement. **Discussion:** Our study uncovers the dynamics of a diabetes team and its members. Healthcare providers work mainly alone, except in hospitals, where they can consult other healthcare providers briefly if necessary. Although collaboration proves to be difficult, all healthcare providers ask for a more intensive interprofessional collaboration. **Conclusion:** In order to improve quality of diabetes care, patient-centered care and the satisfaction of patients, informal caregivers, and healthcare providers, efforts have to be made to facilitate interprofessional collaboration. This can be achieved by sharing information via electronic shared patient records, coordination, overview, local task agreements, simplified legal regulations, and an adjusted financing system.

**Keywords:** home nursing; diabetes team; homecare; interprofessional care

## 1. Introduction

The International Diabetes Federation (IDF) estimated that by 2035, about 600 million people worldwide will suffer from diabetes [1]. The prevalence of diabetes in Belgium is estimated by the IDF at 8% of the adult Belgian population or 1 in 12 adults. This number is predicted to increase to at least 9.6% or 1 in 10 adult Belgians by 2030 [2].

Diabetes is a complex, chronic disease. The optimal treatment of diabetes has been extensively investigated, with a combination of therapeutic patient education, medication, lifestyle and dietary adjustments, and prevention of complications as the gold standard [3,4]. Addressing the non-adherence to diabetes therapy is one of the leading goals of healthcare providers when caring for persons with diabetes. Up to 34% of diabetes patients are

known to be non-adherent to their therapy. Before their diabetic (medication) care is started, patients should be followed up intensively with counseling, health education on the importance of good adherence and on emphasizing the negative consequences of poor adherence [5,6].

However, good diabetes care requires ongoing multidisciplinary collaboration and patient self-management. Efficient use of the available work resources and care providers, ranging from primary to secondary and more specialist care settings, is necessary to deal with the upcoming diabetes epidemic. In its Medical Standards for Diabetes Care (2018), the American Diabetes Association states that diabetes care is best provided by a multidisciplinary team. This guideline refers to a therapeutic team, a healthcare team, and a care team. The guideline also states that the treatment team must include a doctor, a nursing diabetes educator, a dietician, a social worker, and the patient [4]. The Canadian Diabetes Association Clinical Practice Guideline (2013) refers to an interprofessional or multiprofessional team around a person with diabetes, consisting of healthcare providers who improve the outcome of persons with diabetes, such as nurses, dieticians, pharmacists, and psychologists. Here as well, it is suggested that a person with diabetes should be part of this diabetes team to stimulate the self-care [7]. A qualitative study, from the perspective of patients, shows that broad, multidisciplinary, and well-coordinated teams, guided by patient experience, can improve diabetes management. Disciplines mentioned as being important included general practitioners, specialist endocrinologists, dieticians, nurses and nurse practitioners with diabetes expertise, social workers/case managers (to help navigate socioeconomic barriers to care), and physical therapists or others who can help promote physical activity [8].

In Belgium, the organization of diabetes care is a complex matter. Persons with type 1 diabetes are cared for in diabetes centers in secondary and specialist care settings. They receive intensive specialized care by an endocrinologist, nurse diabetes educator, dietician (diabetes educator), podiatrist, and a psychologist. The general practitioner is not often involved in their diabetes care. Persons with type 2 diabetes have multiple options. If the problems remain limited, only the general practitioner and the dietician in primary care are involved. In the case that the person's health worsens, they can step into the pre-care trajectory or the care trajectory for persons with type 2 diabetes. In certain conditions, persons can consult a professional with additional competence as diabetes educator: a dietician, nurse (diabetes educator), dietician (diabetes educator), and a podiatrist (diabetes educator). Only persons with type 2 diabetes—requiring injection therapy—can be integrated in a care trajectory for diabetes.

There is a lack of research with regard to the functioning of—and the roles and tasks in—the diabetes team: which disciplines need to be involved, how to establish continuity in the collaboration between healthcare providers and with persons with diabetes and their family, what is the foundation of a good diabetes team, etc. When do you decide that the quality of the care delivered by a team is 'good enough' and what does 'good enough' mean? Using a single threshold to which all practices should adhere is not a good way to decide if the diabetes care is good enough. It is important for a team to involve the patient and his family as informed full participants: the patient and his family are aware of all the possible outcomes and the team knows how they perceive, prefer, and value these outcomes [9]. The purpose of this study is to explore the characteristics, the roles and tasks in, and the level of collaboration in a diabetes team in primary care in order to improve quality of diabetes care, patient self-management and the satisfaction of patients, their informal caregivers, and healthcare providers with the provided care. The long-term aim is to identify team approaches that are successful to address the unique expertise of all team members in the diabetes team that surrounds persons with diabetes and their families.

## 2. Methods

A descriptive, explorative qualitative study was conducted using the technique of focus groups and in-depth interviews.

## 2.1. Setting and Sample

Purposive sampling, combined with theoretical sampling, was used to recruit persons with diabetes, informal caregivers, and healthcare providers, such as (home) nurses, general practitioners, specialists, diabetes educators, dieticians, psychologists, pharmacists, and podiatrists, throughout Flanders and Brussels to aim for the involvement of all disciplines in diabetes care. Both university hospitals and regional hospitals were approached to participate in the study. The home nurses in this study were employees of an organization for home nursing or were (independent) self-employed home nurses.

Eligible persons with diabetes and informal caregivers were: (a) (caring for) people suffering from type 1 or type 2 diabetes, (b) age $\geq$ 16 years, and (c) willing to share their thoughts and experiences and to participate in a focus group. Patients with severe psychiatric disorders and cognitive impairment were excluded. For healthcare providers the inclusion criteria were: (a) working within direct diabetes care at home or in a hospital setting and (b) willing to share their experiences.

## 2.2. Ethical Consideration

Permission for this research was obtained from the Commission for Medical Ethics of Ghent University Hospital (Approval number B670201836703). Before the start of the study, participants were sufficiently informed about the study, its objectives, potential benefits, and risks. They also received written information and agreed via an informed consent form. All participants could withdraw their consent at any time during this study without any impact on their diabetes care.

## 2.3. Data Collection and Procedure

Four focus groups were planned with healthcare providers and two focus groups with patients and/or informal caregivers. During the recruitment phase, the general practitioners and specialists stated that it was difficult for them to schedule time in their agendas to participate in a focus group. Therefore, two additional in-depth interviews were organized, one with a general practitioner and one with an endocrinologist. Semi-structured interview guides were developed for both the focus groups and in-depth interviews, covering topics about team composition, division of tasks and roles, collaboration, and barriers and facilitators of collaboration.

All six focus groups and two in-depth interviews were organized over a period of 2.5 months in 2018. The size of the focus groups ranged from 5 to 9 participants and lasted 60–90 min. For the in-depth interviews, one researcher visited the doctors' practice. All focus groups and in-depth interviews were audio-taped after consent of the participants. These recordings were transcribed verbatim and deleted after the transcription of the interview.

## 2.4. Data Analysis

The data analysis of this study was an iterative process based on the Qualitative Analysis Guide of Leuven (QUAGOL) [10]. The first part of the analysis focused on the preparation of the coding process: reading and re-reading the transcripts, extracting important fragments, writing a conceptual and narrative report of each focus group/in-depth interview, as well as a conceptual scheme that resulted in a global conceptual scheme. In the second part the ideas in the global conceptual scheme were used to develop a code list. Meaningful fragments were assigned to these codes by means of the QSR NVivo11.0 software program [11]. Each step in the data analysis was discussed with the research team (EVN, ID, KS, LP, KDV), using the constant comparative method (forward–backward dynamic).

## 3. Results

A total of 11 patients ($n$ = 11) and four informal caregivers ($n$ = 4) participated in two focus groups: six men, seven women, and two couples (each couple counting as

one participant). Participants had an average age of 66.8 years. Eight participants were diagnosed with type 2 diabetes and three participants were diagnosed with type 1 diabetes. Participants suffered from diabetes for an average of 14 years.

The sample of healthcare providers consisted of 34 individuals (*n* = 34), of whom 29 were women. The average age of the healthcare providers was 50 years. The nursing diabetes educators were most represented in the study (*n* = 18). All other disciplines (dietician diabetes educator, general practitioner, pharmacist, psychologist, internist, endocrinologist) were represented by two or fewer participants. The healthcare providers had an average of 17.9 years of experience in diabetes care.

The results revealed four major themes with regard to patient-centered care, the diabetes team, collaboration, and points of improvement (bottlenecks).

### 3.1. Patient-Centered Care

All the participants in this study emphasized the importance of the central position of the person with diabetes in the diabetes team: the person with diabetes is in charge of his own care, since he is the expert by experience. However, the translation of this 'desire/wish' into practice seemed not that easy. From the perspective of the healthcare providers in this study, it was difficult to give the management of care to the person with diabetes and his family because of the fear of a lack of knowledge and sufficient adherence with the therapy. From the perspective of the persons with diabetes and informal caregivers, they felt a lack of recognition of their role as 'expert by experience' and at the same time they were uncertain about their level of knowledge in order to be fully in charge of their diabetes care.

> *Healthcare provider: "Yes, I think we must be able to explain the central role of the patient. You can also divide your patients into groups. You have those who are motivated and who are indeed going to do that correctly. But I also have a lot of patients who are not motivated and if I would ask "When do you want to come back?", their answer is within five years. Or never. Sometimes it is necessary to say: "Look, you will come back to me within three months with those specific goals to be obtained. We will take control."*

> *Informal caregiver: "It is actually very important to notice that in first instance you have to do this yourself. You have to take control of the entire process. You have to understand what's happening and check it all yourself. We even see that professionals really make mistakes against that. For example, at the time a surgery was required, we entered into the hospital. An entry form mentioned which type of medication was taken, which diseases the patient has and so on. The nurse filled in the form. She thanks us and prepared an intravenous perfusion in advance of the surgery. That perfusion turned out to be full of glucose. It was the same nurse who had asked us which type of diseases she has. So bottom line, you have to check it yourself."*

### 3.2. Diabetes Team

The participants strongly endorsed the idea that persons with diabetes need to be surrounded by a team. However, there are still persons who are followed only by one healthcare provider, for example only the endocrinologist or general practitioner. Reasons for not consulting professionals were not feeling the need to involve or consult other healthcare providers as the diabetes is under control or because the person with diabetes does not want to allow other healthcare providers in his/her diabetes care.

Where there is a team, a team approach is initiated according to the needs and preferences of the person with diabetes and is based on the collaboration preferences of the healthcare providers. They preferentially refer to healthcare providers they know and work with on a regular basis. Consequently, the composition of the diabetes team can change throughout the disease process. The participants describe a core team of disciplines that are often involved, such as the patient and his informal caregiver, a general practitioner and/or endocrinologist, a nurse diabetes educator, and a dietician (diabetes educator). Depending on the needs of the person with diabetes, other healthcare providers are involved ad hoc. Table 1 specifies all the disciplines that can be involved according to the participants.

**Table 1.** Disciplines involved in diabetes care.

| The Core Team | Ad Hoc Healthcare Providers |
|---|---|
| • Medical specialties<br>   ○ General practitioner<br>   ○ Endocrinologist<br>   ○ Ophthalmologist<br>• Nurse diabetes educator<br>• Dietician (diabetes educator)<br>• Informal caregivers<br>• Patient | • Other medical specialties (for example cardiologist, nephrologist, etc.)<br>• General nurses<br>• Pharmacist<br>• Podiatrist (diabetes educator)<br>• Burn-out coach<br>• Physiotherapist<br>• Psychologist<br>• Tobaccologist<br>• Family care service<br>• Care pathway promotor |

Most of the participants described a clear task and role division of all team members. Tasks are assigned according to the presence of specific knowledge and experience and the level of training/education. All professional members of the core team independently perform a number of common tasks such as foot control, therapeutic patient education, and referral and training/coaching of colleague healthcare providers.

*Person with diabetes: "My coach can explain the disease techniques very well. And she also makes me responsible by acting as a coach, a diabetes specialist, I mean a nurse . . . They teach you a lot about your illness in such a way that you understand that mechanism well. If you understand that correctly, you will also take care of yourself much better."*

*Healthcare provider: "Innovations in diabetes care are following rapidly. Endocrinologists have a great deal of experience in diabetes care and their task is to share information about new treatments with colleagues in primary care so that they are also made aware of this."*

The healthcare providers were able to describe their own role in detail but appeared to have a limited insight into the tasks of colleagues from other disciplines.

*Healthcare provider: "I definitely think that the collaboration with the podiatrists could be improved because we do not know each other well enough, do not know what a podiatrist can and cannot do and for what problems they can be involved. Maybe we could refer people easier once we see what they do exactly? There are not that many podiatrists either. I don't even know if there are podiatrists who specifically deal with diabetes or if that's part of the work of all podiatrists. I don't even know that."*

The nurse diabetes educator has a very broad role and seems to be the first point of contact in case of problems for patients. He/she explains many specific situations such as illness and travelling and teaches techniques. He/she performs a bridge between a person with diabetes and the other team members. Table 2 shows an overview of all the tasks and roles of the different team members as provided by the participants.

*3.3. Collaboration*

The participants clearly described the conditions for collaboration within diabetes care: knowing each other, building a relationship of trust, and sharing the same vision on the diabetes care. Healthcare providers mentioned that they collaborate to promote and ensure adherence of a person with diabetes by providing the same information and advice to that person.

*Healthcare provider: "Diabetes care is actually a multidisciplinary care. Diabetes is teamwork. The doctor diagnosed the patient when this was not yet done. He suggests a therapy schedule. The task is to explain the therapy schedule and to motivate the patient to adhere to it. We try to explain it all, knowing that we don't always succeed. But then the next step, and I really see it that way, is if the educator plays the role in further motivation. Explain, but above all motivate and repeat the message that has already been delivered."*

**Table 2.** Overview of disciplines and their tasks.

| Disciplines | Members of the Discipline | Tasks | |
|---|---|---|---|
| Patient | | • Therapy adherence<br>• Ultimately responsible/take control of disease him or herself | • Undergo the disease process<br>• Illness insight a motivation |
| Informal caregiver | | • Support (self-care of patient and therapy adherence)<br>• Follow therapeutic patient education<br>• Buying materials and medication at the pharmacy | • Accompanying patients when they have a consultation<br>• Acting in case of a crisis<br>• Managing household |
| Other medical specialties | Endocrinologist | • Diagnosis/clinical examination<br>• Interpretation of lab results<br>• Follow-up<br>• Coordination of care<br>• Prescribe medical therapy | • Searching for complications<br>• Administration<br>• Advice<br>• Referral |
| | General practitioner | • Diagnosis/clinical examination<br>• Interpretation of lab results<br>• Follow-up<br>• Coordination of care for type 2 diabetes<br>• Prescribe medical therapy | • Searching for complications/screening<br>• Administration<br>• Advice<br>• Referral<br>• Acting in a crisis situation |
| | Ophthalmologist | • Screening/examination of the eyes | • Prescribe glasses |
| | Psychiatrist | • Diagnosis and treatment of psychiatric diseases | |
| | Dentist | • Mouth care | |
| | Cardiologist | • Diagnosis and treatment of cardiac complications | |
| Nurses | Diabetes educator | • First point of contact in case of problems<br>• Support coordination (home care)<br>• Choice of glucometer<br>• Reading and interpreting glycemia<br>• Emotional support<br>• Screening for problem inclusive feet<br>• Answering questions<br>• Advise extra muros patients (hospital) | • Explaining therapy goals<br>• Remote advice<br>• Administration<br>• Making reports<br>• Act as an intermediary<br>• Lectures<br>• Therapeutic patient education<br>Advise referral |
| | Homecare nurse | • Support patients with technical acts<br>• Screening feet | Administering injections/checking glycemia |
| | Nurse in a general practice | • Supporting the general practitioner<br>• Blood tests<br>• Preparing consult with general practitioner | |
| Dietician (diabetes educator) | | • Nutrition and exercise<br>• Same tasks nurse diabetes educator, except for injection and self-monitoring techniques and first point of contact with problems | • Therapeutic patient education<br>Advise for referral |
| Pharmacist | | • Guiding and follow-up on medication/therapy adherence<br>• Therapeutic patient education | Delivery of material injection/self-inspection and delivery of material for self-control for patients with a care pathway |

**Table 2.** *Cont.*

| Disciplines | Members of the Discipline | Tasks | |
|---|---|---|---|
| Burn-out coach | • Support for team members | | |
| Services home care | • Identifying problems | | |
| Physical therapist | • Motivation to move/doing exercises | | |
| Social work | • Inform about rights | • Identifying problems | |
| Podiatrist (diabetes educator) | • Foot care<br>• Prevention<br>• Checking shoes<br>• Wound care in case of foot wounds | • Making arch supports on prescription<br>• Therapeutic patient education | |
| Psychologist | • Guidance and treatment for psychological comorbidities | | |
| Secretariat | • Answering telephone<br>• Co-coordination of therapeutic patient education (home care)/appointments (hospital setting) | • Support administration (hospital setting)<br>• Listening to patients' stories | |
| Tobaccologist | • Assistance with smoking cessation | | |
| Care pathway promotor | • Communication about regulations (home care)<br>• Informing healthcare professionals | | |

Collaboration is situated on two levels. At the micro level, there is direct collaboration for a specific person with diabetes. Participants endorsed that interprofessional collaboration is the ultimate goal, but that it was not yet fully established. At this moment, the collaboration is multidisciplinary in which healthcare providers follow each other in the care process without knowing much about each other.

*Healthcare provider: "Every healthcare provider has his own task. For example, the pharmacist provides the patient with the blood glucose monitoring meter. But when the meter needs to be installed, the patient calls the nurse diabetes educator. Also when there are technical problems with the glucometer."*

The healthcare providers in this study call each other in case of problems, exchange reports or results of lab tests, either on paper or electronically, or discuss problems face-to-face in a hospital setting. From the perspective of the person with diabetes, no micro-level collaboration is noticed, and they express the assumption that this professional communication is going ahead electronically.

*Healthcare provider: "The collaboration with the second line is very good. I write a referral letter and I always receive a report back. You can always e-mail or call, that goes very well. Uh, with the diabetes educator I am cooperating very well, with the pharmacist also. I have no complaints about the collaboration."*

*Person with diabetes: "When I consult for diabetes in the hospital I see the dietician, the endocrinologist and the nurse diabetes educator. But I see them one after another. I think they work well together but I think it is important that they can sit together and discuss problems. I miss that right now."*

At meso level, collaboration takes the form of a consultation with colleagues, interprofessional training sessions, discussion meetings, etc. These moments are a suitable moment to make task agreements about the division of roles and tasks between the different disciplines, to get to know each other, and to share visions of good diabetes care. However, there is also still room for intensifying collaboration at meso level.

*3.4. Bottlenecks*

All participants mentioned a lot of bottlenecks, which are mainly related to insufficient collaboration as a real team in the support of a person with diabetes. From the perspective of the healthcare providers, this concerns the lack of clear insight in each other's tasks, insufficient referral, fragmented care, the insufficient coordination of care, the lack of a common shared file for information exchange, and the lack of therapy adherence and of responsibility for one's own health status by the person with diabetes and/or the informal caregivers.

> *Healthcare provider: "Things are not yet going as they should go. That is of course problematic if not every healthcare provider is for example looking at one and the same medication system. Which is now even a problem between general practitioners and pharmacists sometimes, things are still going wrong in the first line and between the first line and the hospitals."*

Persons with diabetes indicated that they mainly feel insufficiently recognized as an expert by experience. Persons with diabetes indicate that they frequently collide with long waiting times before they can, for example, consult an ophthalmologist and there is a lack of care coordination. As a result, patients often notice that care is fragmented.

The knowledge of persons with diabetes and healthcare providers also appears a point of improvement. This concerns the lack of up-to-date knowledge about diabetes and the regulations by the healthcare providers, and the lack of insight into the disease by patients and their informal caregivers.

> *Healthcare provider: "I have no bad intentions but pharmacy assistants do a lot or overestimate themselves. They advise their patients wrongly on how to take medication, even if there was a correct prescription."*

> *Healthcare provider: "They must ensure that you have one transparent shared patient record. But that's never going to be the case because everyone has his/her own record and working method. So that will never happen."*

The complex and unclear regulations appear to be a major bottleneck for all the participants in this study. These regulations are insufficiently aligned at the individual needs and requirements of persons with diabetes. New regulations or amendments to existing regulations are reaching healthcare providers in the field very slowly. In addition, healthcare providers indicate problems with the reimbursement. These reimbursements appear to be insufficient to take into account the time it takes to care for or educate a patient with diabetes.

Persons with diabetes experience the regulations and conditions with regard to the reimbursement of the care by healthcare providers as a problem and an obstacle to involve certain disciplines, such as a dietician.

> *Healthcare provider: "I also see people who had in the past a convention but now they have a care pathway. They receive now 140 strips and when I ask how frequent they check their glycemia, they maybe will tell me 60 times. When I ask what they did with the left overs, well they tell me: "oh my wife has prediabetes and she uses the left overs.""*

## 4. Discussion

This research provides a first unique insight into the collaboration within the diabetes care in Flanders, Belgium, and their dynamics. Our study confirms that 'the' diabetes team does not exist and this team is composed in a case-oriented and ad hoc manner. The composition of the diabetes team can change through time. A reassuring result of our study is that there is no discussion among the healthcare providers regarding what good diabetes care consists of (i.e., lifestyle modifications, a healthy diet, and medication), which is supported by international guidelines [3,4].

Persons with diabetes are still solely monitored by one healthcare provider, often the endocrinologist or general practitioner. This can partly be explained by the fact that the

majority of the participants were persons with type 2 diabetes. Physicians usually feel no need to refer a person with diabetes for more specific follow-up or information, but neither does the person with diabetes. The final responsibility concerning the referral and collaboration with other healthcare providers, ultimately lies with the person with diabetes. Referral of patients usually happens when the healthcare provider is (personally) known and trusted, which also implies a shared vision of good diabetes care. As a result, healthcare providers only refer to one or very few colleagues within another discipline. It should be noted, however, that this type of good practice is not generally used across the whole region of Flanders. There are cases where task agreements are made between disciplines, which provide clarity to healthcare providers in the field and promote collaboration and patient referral and in the extension thereof, improve the quality of diabetes care. These results are in line with previous research stating that decisions about professional collaboration can directly influence a patient's experience. McDonald et al. (2012) stated that interventions with the aim of building personal relationships and establishing agreed rules can improve the amount of trust and respect between healthcare providers, a recommendation that can improve the collaboration between diabetes healthcare providers [12].

In the case of collaboration or referral, healthcare providers follow one after the other with little exchange of information, with the exception of a report or through telephone consultation. Another complicating factor for collaboration within the diabetes team is the absence of a shared electronic file. This implies a lack of information through which healthcare providers can provide optimal support to persons with diabetes. Moreover, it is unclear for them in which system persons with diabetes are cared for and what they are entitled to. Ongoing communication between healthcare providers has been considered crucial by physicians to ensure the best possible care for persons with diabetes and should be incorporated within the diabetes team [13]. Good communication between healthcare providers can improve patient care and efforts should be made to attain this goal. Our results show that all healthcare providers considered ongoing communication between all healthcare providers crucial.

There is a clear division of tasks and roles based on the expertise and previous education of the healthcare provider involved. Each discipline within the diabetes team considers their role important and clear. However, healthcare providers are not informed enough about other healthcare providers' roles. This knowledge deficit may underlie the fact that healthcare providers do not refer persons with diabetes enough. The goal of preventive patient-oriented care is therefore rarely obtained and, in addition, legal regulations and inadequate financing systems complicate profound collaboration. It should also be noted that healthcare providers describe their role more broadly than persons with diabetes and informal caregivers. The role of the nurse diabetes educator is also described broadly as they are able to support the person with diabetes in a more comprehensive way, thanks to the broad education they receive as nurses. Our research shows that nurse diabetes educators aim to fill the gaps within diabetes care, which is confirmed by the Royal College of Nursing [14]. Collaboration has been known to be influenced by disagreements and conflicts over roles and role boundaries [12]. Clarifying the team members' roles and defining the division of tasks may enable the collaboration within the diabetes team.

The bottlenecks mentioned in our study were mainly related to insufficient collaboration as a real (diabetes) team. In a world that is changing rather fast, a transdisciplinary approach could be an answer for the complexity and multidimensionality of a chronic disease, such as diabetes. This concept has been clarified in Van Bewer's concept analysis in 2017 as complex and sophisticated, and with the defining attributes of the transcending of disciplinary boundaries, a sharing of knowledge, skills, and decision making, and a focus on real-world problems [15]. This approach could reduce the possibility of missing important information about diabetes patients and would improve their management and could therefore be of use in diabetes care.

### 4.1. Limitations

The main limitation of this study is that most focus group participants were primarily nursing diabetes educators. Recruitment of participants was time-consuming and often difficult as healthcare providers had difficulty finding the time to participate due to their work activities. On the other hand, we ensured that one representative from different healthcare disciplines took part in the study, for example, a psychologist, podiatrist, and endocrinologist to create representativeness in the study. Healthcare providers worked geographically throughout Flanders, whereas persons with diabetes and their informal caregivers lived in two provinces. This was mainly due to the two locations chosen to organize the interviews with persons with diabetes and their informal caregivers. We believe that, despite the limited number of participants, saturation in data collection was achieved in terms of central concepts. The key messages recurred in every focus group and interview.

The quality of the results was guaranteed as the researcher was supported by a supervisory team with extensive experience within qualitative research and the use of QUAGOL methodology. Each interview and focus group was read and analyzed by at least two researchers, ensuring sufficient research triangulation. Consultation within the research team took place at regular intervals and the results were discussed within the team, adjusted or validated when necessary. Analysis took place using the continuous forward–backward method. This technique involves continuous evaluation throughout the analysis, through feedback of the results of prior interviews or focus groups.

### 4.2. Implications for Practice

This qualitative approach 'unraveled' the importance of a multidisciplinary approach of diabetes care. Each professional has a crucial role in the treatment, adherence, follow-up, and satisfaction of the person with diabetes and their informal caregivers. 'Together' is definitely the key word in diabetes care.

Healthcare providers indicate that in an ideal situation the person with diabetes would co-ordinate his/her disease and care, but factors such as therapy non-compliance complicate their role. The central role of a person with diabetes within the diabetes team should be further addressed and expanded.

There is a great need for a shared electronic file to ensure the most up-to-date and relevant information. This could further improve quality of care and reduce the timely pressure of healthcare providers. This file should be accessible at all times and modifications or remarks made by other healthcare providers should be indicated.

Agreements concerning tasks and roles improve the collaboration between healthcare providers. Until now, this has always remained limited to local initiatives, while there is a need to expand this throughout Flanders and Brussels. It could be useful to determine the added value of these local agreements by means of implementation research. Knowing and trusting other healthcare providers is essential in collaboration. Within local structures, there should be continued attention to the accumulation of knowledge among all involved disciplines through training and education.

At present in Flanders, no healthcare provider has a full overview in which system the person with diabetes is registered and what they are entitled to. Persons with diabetes are often referred to a healthcare provider where conditions of reimbursement are no longer met, resulting in the lack of reimbursement of the healthcare provider. For example, the nurse diabetes educator supports the general practitioner in the primary care with regard to the coordination of care for persons with type 2 diabetes, but this remains without financial compensation. The consequence may be that care is fragmented since continuity cannot be guaranteed. In other words, there is a great need for (reimbursed) coordination and oversight. In addition, there has been reference made to a different, more accurate, financing system in accordance with the actual provided care. Stratifying patients according to risk categories (e.g., a flowchart for treatment care pathways in patients with a high risk of comorbidities such as psychiatric conditions) could be a successful approach to

tackle the high costs of defragmenting care for diabetes patients and organizing trans-/multidisciplinary care. This strategy may improve the use of specialized human resources that could be involved in second-level interventions when needed.

Finally, on a larger scale, there is a demand to clarify and detangle regulation, in particular for type 2 diabetes. This would allow healthcare providers to have a better understanding of the conditions in which care can be provided. In addition, governments should work on disseminating new and modified regulations in a more timely and clear manner. Simpler regulation could also promote collaboration, as it would clarify what can be expected of one another and what can be reimbursed.

## 5. Conclusions

This study unraveled the dynamics within the diabetes team in Flanders, Belgium. Persons with diabetes need to be surrounded by a team. Diabetes care is fragmented whereby the involved healthcare providers have their own specific tasks, linked with their education and experience. They pick up several common tasks, for example, therapeutic patient education and coaching of healthcare providers. Knowing and trusting other healthcare providers is essential for collaboration. However, the ultimate goal of interprofessional collaboration has not been obtained yet. This study also unraveled many bottlenecks such as the fragmented care, the lack of knowledge of healthcare providers, the lack of motivation of the diabetes patients, and the difficult regulations and insufficient reimbursement.

**Author Contributions:** Conceptualization: E.V.N., I.D., E.D.W. and K.D.V.; data curation, E.V.N.; formal analysis, E.V.N., I.D., K.S., L.P. and K.D.V.; investigation, E.V.N., I.D., E.D.W. and K.D.V.; methodology, E.V.N., I.D., K.S., L.P. and K.D.V.; project administration, E.V.N.; software, E.V.N. and K.D.V.; supervision, K.D.V.; validation, K.D.V.; writing—original draft, E.V.N. and I.D.; writing—review and editing, K.S., L.P., E.D.W. and K.D.V. All authors have read and agreed to the published version of the manuscript.

**Funding:** This research received no external funding.

**Acknowledgments:** The authors would like to thank all participants in this study and the Working Group Diabetes NVKVV for their expertise and support through the study process.

**Conflicts of Interest:** The authors declare that there are no conflict of interest and that they did not receive financial support or funding for this study. The study was performed by the authors as employees of the White and Yellow Cross.

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
