# Peer review of "The Diabetes Team Dynamics Unraveled: A Qualitative Study"

_diabetology, doi:10.3390/diabetology3010015_

Round 1

Reviewer 1 Report

It is a very original article that studies all the responsibilities of the health team in diabetes care. However, it has lacked a greater development of the responsibility of patients in their care, perhaps through the formation of associations or self-help clubs in which the educational talks by the different health professionals are more collective and less personalized and could be better evaluated. On the other hand, each health system defines the diabetes care mechanism and although it is true that you describe the experience in a European society, in Latin America the primary health care system sometimes has a doctor, nurse and technician for all the diseases. The same thing happens with electronic medical records, whose difficulty in conversing between professionals continues to be the touchstone.

Reviewer 2 Report

Dear Authors,

I identified minor concerns related to your manuscript. For these reasons, I would recommend the paper with minor revisions.

If you elect to revise and resubmit the paper, please include a letter which details your changes to the paper or your rationale for not making a suggested change.

Introduction

  • There is a large amount of data among non-adherence to treatment in patients with diabetes. The non-adherence to anti-diabetic is an issue that needs to be explain and it is one of the leading goals of the therapeutic èquipe. (DOI:10.1089/bari.2015.0021
  • I suggest you improve the text with these references about patient’ care (DOI: 1016/S0140-6736(11)60767-8;10.1001/jama.2010.810)

Results

  • Line 138: fifteen patients (n=11)? Please correct
  • Table 2: psychological problem comorbidities

Discussion

  • Since you wrote that patients mentioned a lot of bottlenecks (line of 263), I think that you should mention that a “transdisciplinary approach “ of the èquipe would be appropriate rather than an insufficient collaboration. This approach would reduce the possibility of missing important information about patients and would improve his management. (lines 343-354) Explain what transdisciplinary approach is and why it might be useful in diabetes care
  • Moreover you might mention the possibility of introduce the figure of “case manager “ (a nurse for instance) that could coordinate the phases of treatment
  • Finally, you need to strengthen the implication of your study. Is this qualitative approach convenient? Starting from patients point of view, it may represent the first step of a novel perspective of a multidisciplinary staff which aim at improving patients’ satisfaction and adherence? You may look at this paper: DOI:1007/s11695-021-05485-9

Limitation and Conclusion

  • Line 372-379 Since the interviews, team with extensive experience… results in high cost, it would be helpful the stratification of patients according to risk categories. a successful approach could be devising flow charts for treatment care pathways in patients with psychiatric comorbidities or with high risk of comorbidities. This strategy may improve the use of specialized human resources that could be involved in second-level interventions when needed.
